# Surface, Volumetric, and Wetting Properties of Oleic, Linoleic, and Linolenic Acids with Regards to Application of Canola Oil in Diesel Engines

**Anna Zdziennicka [1]** , **Katarzyna Szymczyk [1]**, **Bronisław Jańczuk [1]**, **Rafał Longwic [2],*** and **Przemysław Sander [3]**

1    Department of Interfacial Phenomena, Faculty of Chemistry, Maria Curie-Skłodowska University, Maria Curie-Skłodowska Sq. 3, 20-031 Lublin, Poland
2    Faculty of Mechanical Engineering, Lublin University of Technology, Nadbystrzycka 36, 20-618 Lublin, Poland
3    Department of Vehicles, Lublin University of Technology, Nadbystrzycka 36, 20-618 Lublin, Poland
*    Correspondence: r.longwic@pollub.pl; Tel.: +48-606-785-513

**Abstract:** Oleic, linoleic, and linolenic acids are the main components of canola oil and their physiochemical properties decide on the use of canola oil as fuel for diesel engines. Therefore, the measurements of the surface tension of oleic, linoleic, and linolenic acids being the components of the canola oil, as well as their contact angles on the polytetrafluoroethylene (PTFE), poly(methyl methacrylate) (PMMA), and engine valve, were made. Additionally, the surface tension and contact angle on PTFE, PMMA, and the engine valve of the oleic acid and n-hexane mixtures were measured. On the basis of the obtained results, the components and parameters of oleic, linoleic, and linolenic acids' surface tension were determined and compared to those of the canola oil. Next, applying the components and parameters of these acids, their adhesion work to PTFE, PMMA, and the engine valve was calculated by means of various methods.

**Keywords:** combustion; alternative fuel; canola oil; diesel engine; diesel fuel; n-hexane

## 1. Introduction

Canola oil has very wide applications, particularly in the food and pharmaceutical industries, in the production of biofuels, cosmetics, cleaning agents, lubricants, bitumen emulsions, glue production, and as an ingredient for the production of animal feed [1]. In particular, this oil is frequently used as an additive to improve lubricity of petroleum fuels, owing to its good lubrication properties [2–26]. These properties are correlated, among others, with the saturation and hydroxylation of canola oil components. It was observed that an increase in unsaturation causes a lubricity increase. However, recently, intensive research has been carried out in relation to the use of canola oil not only as an additive to petroleum fuels, but as fuel for the diesel engine [27–33]. Among others, in a review article, Ge et al. [33] stated that canola oil can be used as a good alternative fuel in diesel engines without engine modification. Górski et al. [27] also proved that the common rail diesel engines can run well using canola oil without engine modification. However, other investigators [28–32] suggest that canola oil can be used as fuel for diesel engines in the presence of some additives. The use of oil as a biofuel can be favorable on the one hand, but its influence on the environment (closed circulation of carbon dioxide in the atmosphere) should be taken into account. On the other hand, this can be associated with improvement of agricultural economy. Modern diesel engines adapted for the combustion of petroleum-derived fuels cannot be supplied with natural canola oil. The diesel engine is designed in such a way that combustion starts and course proceeds in parts of the combustion chamber at specified

temperatures. In diesel engines, if the fuel acquires physicochemical properties significantly different from the diesel fuel, the process of combustible mixture formation is disturbed [34]. The most important properties determining the practical use of canola oil include viscosity, density, and surface tension. In addition, wetting of various parts of the engine during its operation with the use of biofuels cannot be neglected either. For better understanding of these physicochemical properties of canola oil, at first it is necessary to determine them for the most important canola oil components [34]. Modern varieties of rape allow to obtain oil, the main components of which are unsaturated fatty acids such as oleic, linoleic, and linolenic acids. Their content in canola oil exceeds 90% [35]. In the literature, it is difficult to find the surface tension components of these acids and their wetting properties. Thus, the purpose of the presented paper was to measure the surface tension and contact angle of unsaturated fatty acids on the polytetrafluoroethylene (PTFE), poly (methyl methacrylate) (PMMA), and engine valve surface. PTFE not only has wide practical application, but is treated as a model hydrophobic solid whose surface tension results only from the Lifshitz-van der Waals intermolecular interactions. From this point of view, it is used for determination of apolar and polar components of liquid or solution surface tension. In turn, PMMA is treated as a monopolar solid whose surface tension results from the Lifshitz-van der Waals intermolecular interactions too, but it can interact with the adherent medium by the Lewis acid-base intermolecular interactions and is used for determination of electron-acceptor and electron-donor parameters of acid-base components of liquids and solutions surface tension. Due to the acid high surface tension and weak wetting properties of the above-mentioned acids, it is necessary to improve them with different additives. It seems that n-hexane should be the most proper of them. Therefore, the measurements of the contact angle on PTFE and the engine valve for oleic acid, whose content in the oil is the highest, with the addition of n-hexane were performed.

## 2. Materials and Methods

### 2.1. Materials

Oleic (Sigma-Aldrich, Saint Louis, MO, USA), linoleic (POCH, Gliwice, Poland), and linolenic acids (Sigma-Aldrich, Saint Louis, MO, USA) were used for the surface tension and contact angle on the polytetrafluoroethylene (PTFE), poly(methyl methacrylate) (PMMA), and engine valve surface measurements. The n-hexane (Sigma-Aldrich) was used as an additive to oleic acid for measurements of the surface tension and contact angle on PTFE and the engine valve of the oleic acid-hexane mixture.

The polymers were obtained from Mega-Tech, Poland, and the engine valve was prepared by removing combustion products and their subsequent polishing. The cleaning procedure of the solids used for the contact angle measurements was described earlier [34].

### 2.2. Methods

The equilibrium surface tension ($\gamma_{LV}$) of oleic, linoleic, and linolenic acids, as well as the oleic acid mixture with n-hexane (5%, 10%, 15% and 20% v/w), was measured by the Krüss K9 tensiometer according to the platinum ring detachment method (du Nouy's method). Before the surface tension measurements, the tensiometer was calibrated using water ($\gamma_{LV}$ = 72.8 mN/m at 293 K) and methanol ($\gamma_{LV}$ = 22.5 mN/m at 293 K). The procedure of the surface tension measurements was described earlier [34].

The measurements of the advancing contact angles of oleic, linoleic, and linolenic acids, as well asthe oleic acid mixture with n-hexane (5%, 10%, 15% and 20% v/w) on PTFE, PMMA and the engine valve were conducted using the sessile drop method. The drop volume was equal to 6 µL. The apparatus and procedure of the contact angle measurements were described earlier [34]. For each system, the contact angles were measured for 30 drops of liquids. The standard deviation of the contact angle value did not exceed 1.5 degrees.

## 3. Results and Discussion

The practical use of rapeseed oil is closely related to its surface, wetting, and adhesive properties. The canola oil includes a whole range of different types of chemical compounds. However, its quality is determined by the presence of unsaturated fatty acids and, above all, oleic, linoleic, and linolenic acids. It is difficult to find their surface, wetting, and adhesive properties in the literature.

### 3.1. Surface Tension of Unsaturated Fatty Acids

The average measured value of oleic acid surface tension is equal to 31.92 mN/m (Figure 1). This value is slightly lower than that reported in the literature (32.79 mN/m) [36] and the surface tension of canola oil [34]. The surface tension of linoleic and linolenic acids was measured and determined from the contact angle on the PTFE surface (Figure 1).

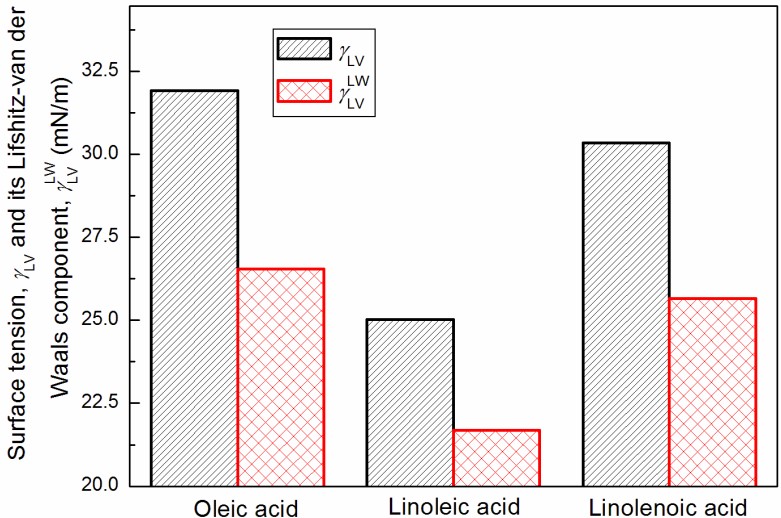

**Figure 1.** The values of the surface tension ($\gamma_{LV}$) and the Lifshitz-van der Waals component of this tension. ($\gamma_{LV}^{LW}$) for oleic, linoleic, and linolenic acids.

The contact angle is closely related to the surface tension of the liquid and solid, as well as solid-liquid interface tension. The relationship between these values is presented by the Young equation, which has the form [37]:

$$\gamma_{SV} - \gamma_{SL} = \gamma_{LV} \cos \theta \tag{1}$$

where $\gamma_{SV}$, $\gamma_{LV}$ and $\gamma_{SL}$ are the solid surface tension, liquid surface tension, and solid-liquid interface tension, respectively, and $\theta$ is the contact angle in the solid-liquid drop-air system. Based on the Young equation, it is possible to determine the surface tension of the solid and the liquid wetting, depending which surface tension is known. The application of the Young equation in relation to the surface tension of a liquid or solid is possible when the dependence of the solid-liquid interface tension on the surface tension of a solid and a liquid is known. Analyzing the contact angles for many different types of liquids on the surface of commonly used polymers, Neumann et al. [38–40] determined a semi-empirical relationship between these parameters. This dependence has the form [38–40]:

$$\gamma_{SL} = \gamma_{LV} + \gamma_{SV} - 2\sqrt{\gamma_{LV}\gamma_{SV}}\,\exp^{-\beta(\gamma_{LV}-\gamma_{SV})^2} \tag{2}$$

where $\beta$ is the constant independent of the kind of the solid-liquid system. By substituting Equation (2) into Equation (1), one can obtain [38–40]:

$$\cos \theta = -1 + 2\sqrt{\frac{\gamma_{SV}}{\gamma_{LV}}}\,\exp^{-\beta(\gamma_{LV}-\gamma_{SV})^2} \tag{3}$$

Using the value of the surface tension of oleic acid (Figure 1) in Equation (3) and its contact angle on the surface of PTFE (Figure 2), the surface tension of PTFE was calculated. This value is equal to 17.75 mN/m, being lower than the PTFE surface tension determined from the contact angle of the n-alkanes (20.24 mN/m) [41]. On the other hand, this value is similar to that of PTFE surface tension determined by Neumann et al. applying their equation for many polar liquids (18 mN/m) [40]. Assuming the value of PTFE surface tension calculated based on the contact angle of oleic acid and its surface tension, and taking into account the contact angle of linoleic and linolenic acid, their surface tension was calculated from Equation (3) (Figure 1). It appears that the surface tension determined in this way is similar to that measured. The difference does not exceed the accuracy of the measurements and therefore, in Figure 1, the values of the surface tension obtained from the contact angle are shown. As follows from Figure 1, the surface tension of linolenic acid is close to that of oleic acid. The lowest surface tension value is exhibited by linoleic acid. The surface tension values of all tested unsaturated fatty acids are lower than that of canola oil (34.15 mN/m) [34].

Van Oss et al. [42–44] argue that wettability of solids by liquids depends not only on their total surface tension, but also on their components and parameters resulting from different types of intermolecular interactions. The authors divide the surface tension into two components: Lifshitz-van der Waals ($\gamma^{LW}$) and acid-base ($\gamma^{AB}$). On the other hand, the acid-base component is a function of electron-acceptor ($\gamma^+$) and electron-donor ($\gamma^-$) parameters. Taking into account such division of the surface tension, van Oss et al. [42–44] proposed a relationship associating the interface tension with the components and parameters of the surface tension of the contacting phases. This dependence for the solid-liquid system has the form:

$$\gamma_{SL} = \gamma_{LV} + \gamma_{SV} - 2\sqrt{\gamma_{LV}^{LW}\gamma_{SV}^{LW}} + 2\sqrt{\gamma_{LV}^+\gamma_{SV}^-} + 2\sqrt{\gamma_{LV}^-\gamma_{SV}^+} \tag{4}$$

If the liquid contacts apolar solid, whose surface tension results only from the dispersion intermolecular interactions, then from Equations (1) and (4) the following is obtained:

$$\gamma_{LV}^{LW} = \frac{[\gamma_{LV}(\cos\theta + 1))]^2}{4\gamma_{SV}^{LW}} \tag{5}$$

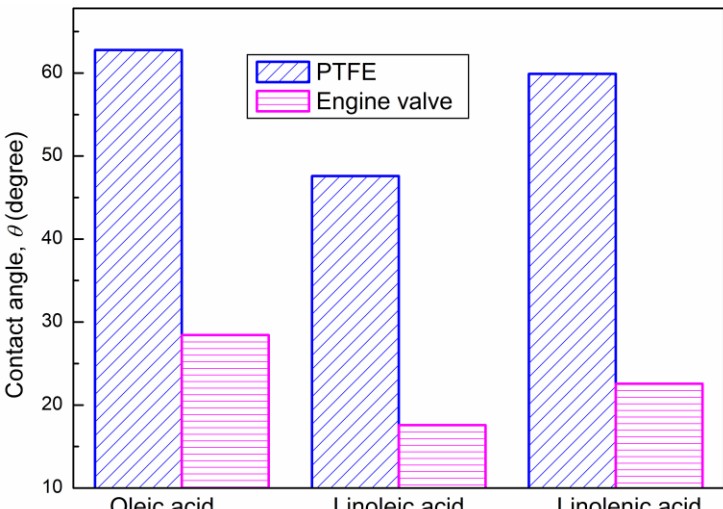

**Figure 2.** Values of the contact angle ($\theta$) for oleic, linoleic, and linolenic acids on the polytetrafluoroethylene (PTFE) and engine valve surface.

By substituting the values of the contact angle (Figure 2) of unsaturated fatty acids measured on the PTFE surface, as well as that of PTFE surface tension which results only from the dispersion

intermolecular interactions [41] into Equation (3), the Lifshitz-van der Waals component of fatty acids was calculated (Figure 1). Next the acid-base component of this tension was determined (Figure 3).

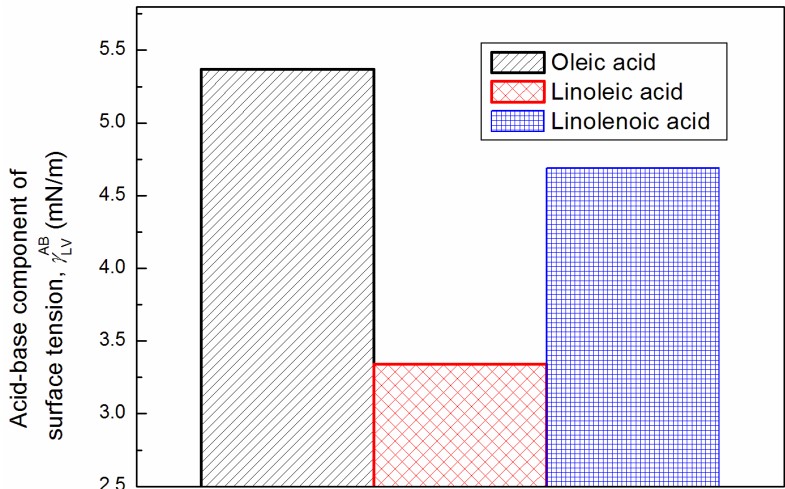

**Figure 3.** The values of the acid-base component ($\gamma_{LV}^{AB}$) of oleic, linoleic, and linolenic acids' surface tension.

Figures 1 and 3 show that the Lifshitz-van der Waals components of oleic and linolenic acids are almost the same, only differing in acid-base components. The values of the Lifshitz-van der Waals and the acid-base components of surface tension of linoleic acid are the lowest of the tested fatty acids. It should be noted that the values of components of the acids' surface tension considerably differ from those of canola oil [34].

From apractical point of view, a decrease of liquids' surface tension by the addition of some substances which have a low surface tension is very advantageous. Therefore, the surface tension of oleic acid and the n-hexane mixture was measured ($C_p$ n-hexane 5% $\gamma_{LV}$ = 21.2 mN/m, 10%—19.1 mN/m, 15%—18.6 mN/m, and 20%—18.3 mN/m). It appear that the measured values of the surface tension are considerably lower than those calculated from the equation for ideal mixtures, which has the form:

$$\gamma_{LV} = \gamma_1 X_1 + \gamma_2 X_2 \tag{6}$$

These differences indicate that n-hexane can adsorb at the oleic acid–water interface. Thus, the composition of the surface region of oleic acid-n-hexane is quite different from that in the bulk phase. As follows from our earlier studies, the surface tension of the mixture can be predicted based on that of components and the surface fraction occupied by the given component of the mixture ($X_1^S, X_2^S$) [45]. If so, the fraction of the interface area occupied by n-hexane ($X_1^S$) can be determined from the following equation:

$$\gamma_1 X_1^S + \gamma_2 (1 - X_1^S) = \gamma_{LV} \tag{7}$$

Knowing the fraction area of n-hexane occupying the oleic acid–air interface, it is possible to determine n-hexane concentration in the surface layer. It is commonly known that the area occupied by n-hexane molecule at the perpendicular orientation is equal to $21\text{Å}^2$ [46]. This corresponds to the n-hexane concentration in the surface region equal to $7.91 \times 10^{-6}$ mol/m². This value should be treated as the limiting concentration of n-hexane at the surface layer ($\Gamma_1^\infty$). Thus:

$$X_1^S = \Gamma_1 / \Gamma_1^\infty \tag{8}$$

Calculated from Equation (8), values of $\Gamma_1$ indicate that at the concentration of n-hexane in the bulk phase equal to 10%, there is almost 90% of n-hexane molecules at the oleic acid–air interface.

This means that the addition of n-hexane to oleic acid causes a considerable decrease in the surface tension because of its high adsorption at the oleic acid–air interface.

### 3.2. Wetting Properties of Unsaturated Fatty Acids

Wetting properties of unsaturated fatty acids were tested on the surface of apolar PTFE, monopolarPMMA, and bipolar part of a car engine [34]. In the case of PTFE, the best wettability is shown by linolenic acid (Figure 2) and the worst by oleic acid. However, any acid spreads completely over the PTFE surface. Over the PMMA surface, the complete spreading is not observed only for oleic acid.

Taking into account the contact angle of oleic acid, the surface tension of PMMA was calculated from Equation (3). Its calculated value is equal to 31.57 mN/m and it is lower than that of PMMA calculated from the contact angle measurements for the model liquids [47]. However, the PMMA surface tension value calculated based on oleic acid data is higher than that of linoleic and linolenic acids. This justifies the total spread of these acids over the PMMA surface. On the other hand, it is possible that during the fatty acids spreading, the film is first created and the surface tension of PMMA covered by this film, only in the case of oleic acids, is lower than its surface tension. In such case, a contact angle higher than zero is formed. This phenomenon can be similar to benzene spreading over the water surface [37]. Similar to PTFE, the best wetting agent of the engine part is linoleic acid, and the worst oleic. The engine part wettability by oleic acid is similar to that of canola oil (Figure 2) [34]. The n-hexane addition to oleic acid improves its wetting properties in relation to the part of the car engine similarly to PTFE (Figure 4). To account for the influence of n-hexane on the wetting properties of oleic acid, the values of the contact angle of oleic acid and the n-hexane mixture were determined from the following equation:

$$\theta_1 X_1 + \theta_2 X_2 = \theta \tag{9}$$

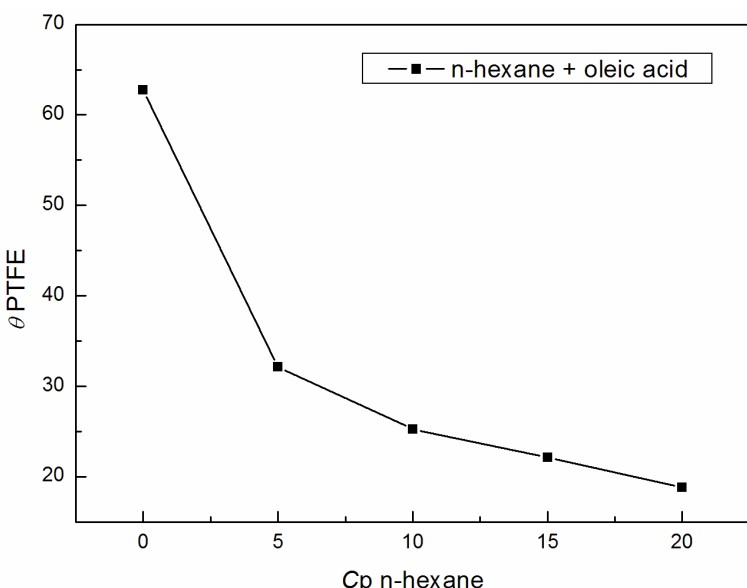

**Figure 4.** A plot of the contact angle ($\theta$) for n-hexane and oleic acid mixtures on the PTFE surface vs. n-hexane percentage concentration ($C_p$).

Indeed, this equation reflects the changes of the contact angle as the function of the ideal mixture composition. Comparing the calculated values of the contact angle to those of the measured ones, a high synergetic effect in the contact angle values is observed. This points out that the measured values of the contact angle at a given composition of oleic acid and the n-hexane mixture are considerably lower than those calculated from Equation (9). This is probably a result of the fact that the concentration of n-hexane at the PTFE–oleic acid interface, similar to the oleic acid–air interface, is higher than in the

bulk phase. The coverage of the PTFE–oleic acid interface by n-hexane molecules can be determined from the following equation:

$$\theta_1 X_1^S + \theta_2 (1 - X_1^S) = \theta \tag{10}$$

The fraction of area occupied by n-hexane calculated from Equation (10) is larger than the mole fraction of n-hexane in the bulk phase, but lower than that of n-hexane at the oleic acid–air interface (Figure 5).

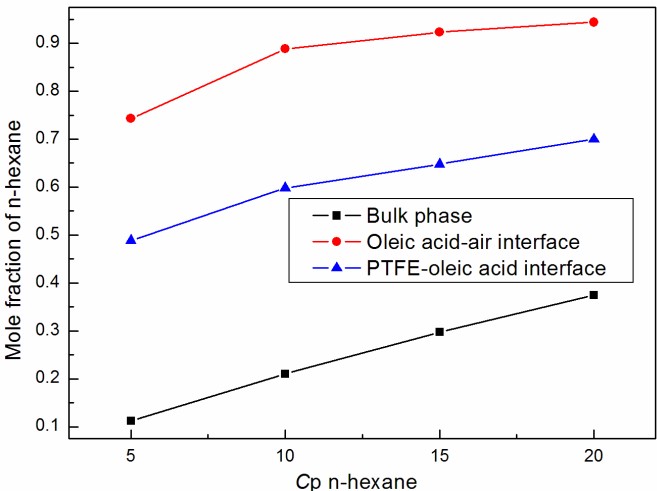

**Figure 5.** The values of n-hexane mole fraction in n-hexane and oleic acid mixture vs. its percentage concentration ($C_p$).

This points out that n-hexane adsorption at the PTFE–air interface is lower than that at the oleic acid–air one. It should be mentioned that the adhesion work of oleic acid to the PTFE surface (46.54 mJ/m$^2$) is higher than that of n-hexane (37.63 mJ/m$^2$). On the other hand, the cohesion work of oleic acid is also higher than that of n-hexane. The balance of adhesion and cohesion work of oleic acid and n-hexane decides about n-hexane adsorption at the PTFE–oleic acid interface. As follows, the difference between oleic acid adhesion to the PTFE surface and its cohesion work is equal to −17.32 mJ/m$^2$, and that for n-hexane is 2.65 mJ/m$^2$.

### 3.3. Adhesion Properties of Unsaturated Fatty Acids

Liquid adhesion to the solid surface plays a very important role in practice. The work of liquid adhesion to the surface of a solid ($W_a$) can be calculated, among others, from the Young-Dupre equation, which is in the form [37]:

$$Wa = \gamma_{LV}(\cos\theta + 1)) \tag{11}$$

The adhesion work values calculated from Equation (11) are shown in Figures 6 and 7. As these figures show, the best adhesion properties are found for oleic acid and the worst for linoleic. Adhesion work values of the tested unsaturated fatty acids could be compared only in the case of PTFE and the engine part, since linoleic and linolenic acids spread over the PMMA surface completely. In this case, one can only conclude that their work of adhesion to the PMMA surface is greater than the cohesion work. Linolenic acid adhesion to the PTFE surface and the engine part is comparable to that of oleic acid. For all studied fatty acids, the work of adhesion to PTFE and the engine part is lower than that of canola oil to these solids [30]. In the case of oleic acid, it was possible to compare its adhesion work to the surface of apolar, monopolar, and bipolar solids (Figure 7). It appears that the adhesion work of oleic acid to the PMMA surface has the largest value, and that to PTFE the lowest one.

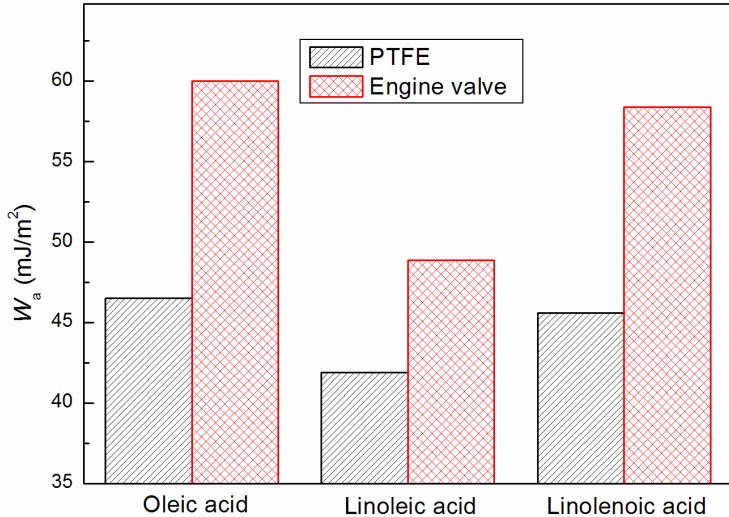

**Figure 6.** The values of the adhesion work ($W_a$) for oleic, linoleic, and linolenic acids to the PTFE and engine valve surface.

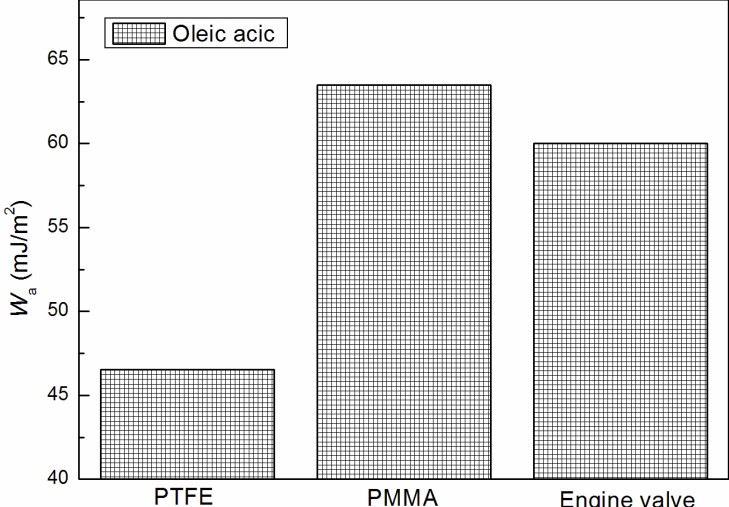

**Figure 7.** The values of the adhesion work ($W_a$) for oleic acid to the PTFE, PMMA, and engine valve surface.

## 4. Conclusions

The results obtained from the measurements of the surface tension show that tension for unsaturated fatty acids is lower than that of canola oil and depends on the kind of acid. The value of the surface tension of oleic acid (31.92 mN/m) is comparable to that which can be found in the literature. The surface tension of linoleic and linolenic acids is equal to 25.02 mN/m and 30.35 mN/m, respectively. Unfortunately, for these acids, it is difficult to find these data in the literature; therefore, it is impossible to make a comparison between the literature values and those measured by us.

Taking into account the measured values of the surface tension and the contact angle of the fatty acids on the PTFE and PMMA surfaces, as well as applying the van Oss and co-workers' concept for the interface tension, it was established that the surface tension of these acids results from the Lifshitz-van der Waals and Lewis acid-base intermolecular interactions. This indicates that the surface tension of fatty acids can be divided for two components—apolar resulting from the Lifshitz van der Waals intermolecular interactions and polar resulting from the Lewis acid-base interactions. However, the apolar component of surface tension is significantly higher than that of the polar one, and represents 83.2%, 86.7%, and 84.6% of the total values of oleic, linoleic, and linolenic acids'

surface tension, respectively. The Lewis acid-base intermolecular interactions are a function of two parameters: Electron-acceptor and electron-donor. For all unsaturated fatty acids, the electron-donor parameter is higher than that of electron-acceptor which is connected with the presence of oxygen in the acids molecules.

The measured values of the contact angle of oleic, linoleic, and linolenic acids on PTFE, PMMA, and the engine valve show that the best wetting properties are exhibited by linoleic acid (PTFE—47.58°; PMMA—0°; and the engine valve—17.57°). However, from the analysis of the Young-Dupre equation, it results that the best adhesion properties are shown by oleic acid ($W_a$ = 46.52, 63.5, and 60.0 mJ/m$^2$ for PTFE, PMMA, and the engine valve, respectively). The values of the contact angle and adhesion work of fatty acids obtained by us are difficult to compare to those in the literature because it is difficult to find such values.

The addition of n-hexane to oleic acid causes a considerable decrease of its surface tension from 31.92 mN/m to 18.3 mN/m in the range of n-hexane concentration from 0 to 20% due to its high adsorption at the oleic acid–water interface. The changes of the surface tension of oleic acid and the n-hexane mixture as a function of the compositionare not linear and synergism in the surface tension reduction is observed. As follows, the oleic acid-n-hexane mixture is not ideal.

The addition of the n-hexane to oleic acid causes a non-linear decrease of the contact angle on the PTFE surface from 62.77° to 18.84°in the range of n-hexane concentration from 0 to 20%. Similarly to the surface tension of the mixture, synergism in the contact angle reduction is observed. The synergism in the surface tension and contact angle reduction by oleic acid and then-hexane mixture results from a higher concentration of n-hexane at the oleic acid–air and PTFE–oleic acid interfaces, rather than in the bulk phase. However, n-hexane adsorption at the oleic acid–air interface is higher than that at the PTFE–oleic acid one.

The decrease of surface tension and improvement of the wetting properties of oleic acid being the main component of canola oil by the addition of n-hexane suggests that it is possible to improve canola oil applied for the diesel engine.

**Author Contributions:** A.Z. and K.S.—signed the experiments, analyzed the experimental data, made figures, and participated in the preparation of the manuscript. B.J.—conceived the concept of the studies, wrote the main part of the manuscript, and supervised the studies. R.L. and P.S.—prepared solids for measurements and participated in the manuscript preparation.

**Funding:** This project/research was financed in the framework of the project Lublin University of Technology-Regional Excellence Initiative, funded by the Polish Ministry of Science and Higher Education (contract no. 030/RID/2018/19).

**Conflicts of Interest:** The authors declare no conflicts of interest.

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
