# Peer review of "Surface, Volumetric, and Wetting Properties of Oleic, Linoleic, and Linolenic Acids with Regards to Application of Canola Oil in Diesel Engines"

_applsci, doi:10.3390/app9173445_

Round 1
Reviewer 1 Report
This study investigated the contact angle and surface tension of oleic, linoleic and linolenic acids along with a mixture of oleic acid and n-hexane on PTFE, PMMA and engine valve. There are several concerns in this study including the improvement of the technical English. Role of physicochemical properties of PTFE and PMMA surfaces have not been discussed properly. It would be interesting if authors could develop and provide the correlation in the surface tension of the binary and tertiary mixtures of the acids in different ratios in order to mimic the use of canola oil for diesel engines.
There are various grammatical and typo errors and manuscript needs careful editing by someone with expertise in technical English.
It is surprising that authors have used the wrong spelling for linolenic acid. Linolenoic acid should be corrected with Linoleic acid throughout the manuscript.
Introduction requires significant improvement in terms of citing proper literature and including details of previously published work on canola oil for the diesel engine.
Authors mentioned on page no. 3, line no. 95: “The surface tension of linoleic and linolenic acid was not measured using directly….” Authors should clearly specify why surface tension not measured using tensiometer?
Why was the contact angle for oleic acid+n-hexane mixture not studied on PMMA and engine oil?
The discussion of the manuscript requires critical improvement and should be written in a paragraph rather than a pointwise format.
Reviewer 2 Report
The quality of the manuscript titled: “Surface, volumetric and wetting properties of the oleic, linoleic and linolenoic acids with regards to application of canola oil in diesel engine” is quite poor. In this sense, I considerer that this work should be not published in the journal Applied Science.
Some of the mistakes and justifications not clearly explained are detailed below:
- I think that linolenoic acid is incorrect. I suppose that the correct acid is “linolenic”. Authors must clarify this concept.
- At the beginning of abstract it is written: “The of the oleic..:”. It is a clear mistake.
- When authors refer to canola oil, is it not clear if they mean oils or esters (biodiesel)?. The use of vegetable oils directly (as fuel) in diesel engine is, nowadays, almost impossible due to the high viscosity of oils that hinders the appropriate atomization of fuel. Authors must clarify is they are talking about canola oil or canola biodiesel. This concept must be clarified.
- Authors do not include the definition of different parameters basic on the development of the work. What is the meaning of engine valve or contact angle? What is the influence of this parameters on the use of this fuel in diesel engines?
- In Figures 1 and 2 the title of y-axis (parameter and units) is not included. In Figure 2, The symbol of contact angle is not written between parentheses.
- In equation 6, the first term of equality is not included.
- The conclusions are presented individually, without any connection between them.
- In line 33, two many references are included in this part of bibliography. If atuhors want to include so many references, they should comment the peculiarities of some of these studies.
Author Response
July 25, 2019
Ms. Lydia Han, M.A.
Assistant Editor,
Dear Lydia Han,
Please find enclosed the revised paper of our article entitled: “SURFACE, VOLUMETRIC AND WETTING PROPERTIES OF THE OLEIC, LINOLEIC AND LINOLENIC ACIDS WITH REGARDS TO APPLICATION OF CANOLA OIL IN DIESEL ENGINE" (applsci-558845) in which we took into account almost all remarks and suggestions of Reviewers and Academic Editor.
We hope that after the revision the paper satisfies the requirements for its publication in Applied Science.
With best regards
Rafał Longwic

Round 2
Reviewer 1 Report
The questions have been appropriately addressed. The modified manuscript should further be corrected for technical English and can later be accepted for publication.
Author Response
Reply to Reviewer comments
The questions have been appropriately addressed. The modified manuscript should be further be corrected for technical English and can later be accepted for publication.
Thank you very much for the time spent on revising our manuscript. We verified all text in terms of linguistic correctness.
With best regards
Rafał Longwic

Reviewer 2 Report
Comments and suggestions are detailed in the attached file

Author Response
Please find enclosed the revised manuscript of our article entitled: “SURFACE, VOLUMETRIC AND WETTING PROPERTIES OF THE OLEIC, LINOLEIC AND LINOLENOIC ACIDS WITH REGARDS TO APPLICATION OF CANOLA OIL IN DIESEL ENGINE" (applsci-558845) in which we took into account almost all remarks and suggestions of Reviewers and Academic Editor.
We hope that after the revision the paper satisfies the requirements for its publication in Applied Science.
With best regards
Rafał Longwic

Round 3
Reviewer 2 Report
Authors have modified almost all the suggestions commented by the reviewer, but English must be revised.
Author Response
The revised manuscript was checked by the English teacher and we hope that it satisfies your requirements. “
Rafał Longwic